# Increased Plasma L-Arginine Levels and L-Arginine/ADMA Ratios after Twelve Weeks of Omega-3 Fatty Acid Supplementation in Amateur Male Endurance Runners

**DOI:** 10.3390/nu14224749

**Published:** 2022-11-10

**Authors:** Zbigniew Jost, Maja Tomczyk, Maciej Chroboczek, Philip C. Calder, Helena L. Fisk, Katarzyna Przewłócka, Jędrzej Antosiewicz

**Affiliations:** 1Department of Biochemistry, Gdansk University of Physical Education and Sport, 80-336 Gdansk, Poland; 2Department of Physiology, Gdansk University of Physical Education and Sport, 80-336 Gdansk, Poland; 3School of Human Development and Health, Faculty of Medicine, University of Southampton, Southampton SO16 6YD, UK; 4NIHR Southampton Biomedical Research Centre, University Hospital Southampton NHS Foundation Trust and University of Southampton, Southampton SO16 6YD, UK; 5Department of Bioenergetics and Exercise Physiology, Medical University of Gdansk, 80-210 Gdansk, Poland

**Keywords:** omega-3 fatty acids, L-arginine, ADMA, nitric oxide, running economy, endurance runners

## Abstract

It is not fully understood how supplementation with omega-3 fatty acids affects the metabolism of amino acids required for the bioavailability/synthesis of NO, i.e., L-arginine (L-arg), asymmetric dimethylarginine (ADMA), their metabolites, and the L-arg/ADMA ratio and their impact on running economy (RE) in runners. Thus, 26 male amateur endurance runners completed a twelve-week study in which they were divided into two supplemented groups: the OMEGA group (*n* = 14; 2234 mg and 916 mg of eicosapentaenoic and docosahexaenoic acid daily) or the MCT group (*n* = 12; 4000 mg of medium-chain triglycerides daily). At the same time, all participants followed an endurance training program. Before and after the 12-week intervention, blood was collected from participants at two time points (at rest and immediately post-exercise) to determine EPA and DHA in red blood cells (RBCs) and plasma levels of L-arg, ADMA, and their metabolites. RBC EPA and DHA significantly increased in the OMEGA group (*p* < 0.001), which was related to the resting increase in L-arg (*p* = 0.001) and in the L-arg/ADMA ratio (*p* = 0.005) with no changes in the MCT group. No differences were found in post-exercise amino acid levels. A total of 12 weeks of omega-3 fatty acid supplementation at a dose of 2234 mg of EPA and 916 mg of DHA daily increased levels of L-arg and the L-arg/ADMA ratio, which indirectly indicates increased bioavailability/NO synthesis. However, these changes were not associated with improved RE in male amateur endurance runners.

## 1. Introduction

Supplementation with omega-3 fatty acids, particularly eicosapentaenoic acid (EPA) and docosahexaenoic acid (DHA), has effects that include, but are not limited to, a reduction in the risk of cardiovascular diseases [1,2], nervous system diseases [3] and metabolic diseases such as diabetes mellitus [4]. Moreover, in healthy, trained and/or untrained subjects, supplementation with omega-3 fatty acids has been shown to enhance muscle function and recovery [5,6]. Evidence for performance improvement in endurance athletes following omega-3 fatty acid supplementation is scarce; however, our recent study showed that 12-week supplementation with omega-3 fatty acids in amateur runners increased the so-called omega-3 index (O3I) (expressed as a sum of % EPA and % DHA levels in red blood cells (RBCs)) which was associated with improved running economy (RE) [7]. Nonetheless, the underlying mechanism appears to be complex and is not fully understood. Among the proposed mechanisms is an increase in the release of nitric oxide (NO) by the vascular endothelium, which is characteristic of, among others, aerobic physical training [8]. This phenomenon is possibly due to the metabolism of L-arginine (L-arg) into L-citrulline via endothelial nitric oxide synthase (eNOS); among the products of this transformation is NO [9]. As a result, there is an increase in cyclic guanosine monophosphate (cGMP), which leads to the relaxation of smooth muscle and vasodilation [10].

On the other hand, the vasodilator effect is antagonized in the presence of asymmetric dimethylarginine (ADMA) in plasma, a competitive inhibitor for eNOS [11,12]. Both ADMA and the second amino acid from the methylarginase family, symmetric dimethylarginine (SDMA) negatively correlate with the bioavailability of NO, although the latter weakly and indirectly inhibits NO synthesis [13]. Increased plasma ADMA and/or SDMA levels are related to an impairment of vascular functions, thus becoming a factor increasing the risk of cardiovascular diseases [14,15]. Previous research suggests the L-arg/ADMA ratio as among the robust tools for assessing vascular endothelial function [16]. Low values of the ratio increase the risk of impaired vascular endothelial function, and therefore enhance the rate of hospitalization and mortality [17]. Decreased levels of L-arg and a lower L-arg/ADMA ratio observed after strenuous exercise may result in reduced ability to synthesize NO [18]. Hence, finding an exogenous modulator of these amino acids seems to be important not only for the sedentary, but also for healthy, physically active people and athletes. Despite the positive effect of supplementation with omega-3 fatty acids on the exercise capacity of endurance athletes [19,20], deficiencies of omega-3 fatty acids are still observed, among others, in the diet of NCAA athletes [21].

Mechanisms responsible for changes in amino acid metabolism following supplementation with omega-3 fatty acids are not comprehensively understood, and the effect on L-arg metabolites and the L-arg/ADMA ratio seems to be crucial in understanding the effect of omega-3 fatty acids among athletes. Thus, the aim of this study was twofold—firstly, to investigate the effect of 12-week supplementation with omega-3 fatty acids on the plasma levels of L-arg, ADMA, the L-arg/ADMA ratio and related metabolites and, secondly, to assess whether the aforementioned markers correlate with RE in male amateur endurance athletes.

## 2. Materials and Methods

### 2.1. Participants

Twenty-six male runners (37 ± 3 years old; 77 ± 9 kg body weight; VO_2peak_: 54.2 ± 6 mL*kg^−1^*min^−1^) completed a randomized controlled trial, approved by the Bioethical Committee of Regional Medical Society in Gdańsk (NKBBN/628/2019) and conducted according to the Declaration of Helsinki.

### 2.2. Study Design

This study was part of a larger research project with details outlined elsewhere [7], and characteristics of the participants are shown in Table 1. Briefly, participants were randomly assigned to one of two groups with the final characteristics as follows: OMEGA (age: 37 ± 3 years; body weight: 76 ± 11 kg; VO_2peak_: 53.8 ± 5 mL*kg^−1^*min^−1^) or medium-chain triglycerides (MCT) (age: 37 ± 4 years; body weight: 78 ± 8 kg; VO_2peak_: 54.7 ± 7 mL*kg^−1^*min^−1^). All participants completed a 12-week programme that included 4 training sessions per week (3 running sessions + 1 core strengthening session). The training structure was based on the ventilatory threshold (VT) and ventilatory anaerobic threshold (VAT) method with corresponding three heart rate (HR) zones: [Z1: ≤HR@VT1 + 5 bpm; Z2: (>HR@VT1 + 5 bpm) to (≤HR@VAT-5 bpm); Z3: >HR@VAT-5 bpm]. Simultaneously, participants ingested 4 capsules per day providing a total of 2234 mg of EPA + 916 mg of DHA (OMEGA group) or 4000 mg of MCTs (MCT group). Before and after the 12-week period, VO_2peak_ during an incremental treadmill test was measured on a motorized treadmill (h/p Cosmos, Saturn, Germany) and blood samples were taken twice: before starting and immediately after finishing the test. The test consisted of a few stages: first, participants walked for 5 min at 5 km/h speed and with a 1.5% incline as a warm-up. Second, the treadmill belt was accelerated starting from 8 km/h by 1 km/h per stage up to 12 km/h with every next stage duration of 3 min. Then, the incline of the treadmill was increased to 5%, 10% and 15% at 12 km/h speed until volitional exhaustion. During both tests, heart rate (HR) was monitored (Polar RS400, Kempele, Finland). Additionally, oxygen uptake (VO_2_), carbon dioxide output (VCO_2_), minute ventilation (Ve) and respiratory exchange ratio (RER) were continuously measured using a breath-by-breath analyzer (Oxycon Pro, Jaeger, Hoechberg, Germany). VO_2peak_ was obtained as the highest 30 s mean value recorded during the test. RE was measured as an oxygen cost from last 50 s as previously described [22] with slight modifications accordingly to Tomczyk et al., 2022 [7].

### 2.3. Sample Collection

Blood samples were collected into 4 mL sodium citrate vacutainer tubes and centrifuged at 4 °C (4000× *g* for 10 min). After centrifugation, plasma and RBCs were collected with a disposable Pasteur pipette and transferred into separate Eppendorf probes and stored in a −80 °C freezer until further analysis.

### 2.4. Fatty Acid Analysis

Concentrations of EPA and DHA in red blood cells (RBCs) were measured using gas chromatography [23]. Briefly, RBC lipids were extracted into chloroform methanol and fatty acid methyl esters (representing the RBC fatty acids) were formed by heating the lipid extract with methanolic sulphuric acid. The fatty acid methyl esters were separated by gas chromatography on a Hewlett Packard 6890 gas chromatograph fitted with a BPX-70 column. Fatty acid methyl esters were identified by comparison with run times of authentic standards. Fatty acids are expressed as a % of total fatty acids present.

### 2.5. Amino Acid Assessment

Determinations of plasma L-arginine, ornithine, L-citrulline, DMA, ADMA and SDMA concentrations were performed using high-performance liquid chromatography with tandem mass spectrometry (LC-MS/MS) with prior protein precipitation and derivatization. To 50 µL of plasma, 200 µL of protein precipitation reagent was added (mixture of internal standards in water and methanol, 20:80). The sample was stirred for 15 min (1100× *g* rpm) and centrifuged (3000× *g* rpm, 10 min). A volume of 10 µL of supernatant was transferred to a new insert vial and subjected to AccQ-Tag (Waters Co, Milford, MA, USA) derivatization in accordance with the manufacturer’s recommendations. After derivatization, samples were diluted 1:1 with ultrapure water and subjected to LC-MS/MS analysis accordingly to Carling et al. [24] with slight modifications.

### 2.6. Statistical Analysis

Statistical analysis was performed using GraphPad Prism 7. Each variable was subjected to normal distribution analysis using the Shapiro–Wilk test. Arithmetic means, standard deviation and significance levels were calculated. When the distribution of the variable was normal, the paired *t*-test was used, while when the distribution was not normal the non-parametric Wilcoxon test was used. Then, two-way analysis of variance (ANOVA) with repeated measures to investigate the significance of differences between groups and time was used. Significant main effects were further analyzed using the Sidak post hoc test. Correlations between variables were evaluated using the Spearman correlation coefficient. Significance for all analyses was assumed at *p* < 0.05.

## 3. Results

### 3.1. Omega-3 Polyunsaturated Fatty Acids in RBCs

Baseline levels of EPA and DHA and the O3I did not differ between the two groups (OMEGA group: 1.1% EPA, 4.7% DHA, 5.8% O3I; MCT group: 1.2% EPA, 4.4% DHA, 5.6% O3I, all *p* > 0.999). Post-intervention values of EPA, DHA and O3I increased in the OMEGA group to 4.9% EPA, 6.7% DHA, 11.6% O3I (all *p* < 0.001). Changes were not observed in the MCT group (1.2% EPA, *p* > 0.999; 4.7% DHA, *p* = 0.551; 5.8% O3I, *p* > 0.999).

### 3.2. Plasma L-arginine and Its Metabolites at Resting Conditions

The plasma levels of L-arg and its metabolites for both groups at rest are provided in Table 2 and Figure 1. For L-arg, a statistically significant increase was noted in the OMEGA group (*p* = 0.001), while there was no change (*p* = 0.109) in the MCT group after 12 weeks of supplementation. The level of ornithine was significantly decreased from pre to post in both groups (*p* < 0.001 and *p* = 0.007 for the OMEGA and MCT groups, respectively). Additionally, the L-arg/ADMA ratio was increased in the OMEGA group from pre to post (*p* = 0.005), while there was no change in the MCT group (*p* = 0.077).

### 3.3. Plasma L-arginine and Its Metabolites Post-Exercise

The post-exercise plasma levels of L-arg and its metabolites for both groups are provided in Table 3 and Figure 2. For L-arg, a statistically significant change was observed in both groups after 12 weeks of supplementation (*p* < 0.001 and *p* = 0.016 for the OMEGA and MCT groups, respectively). Additionally, change in the L-arg/ADMA ratio was significant for both groups (*p* < 0.001 and *p* = 0.021 for the OMEGA and MCT groups, respectively). However, there were no differences between the OMEGA and MCT groups in post-exercise levels.

### 3.4. Plasma L-arginine, the L-arg/ADMA Ratio and Running Economy

The correlations between plasma L-arg, the L-arg/ADMA ratio and RE are provided in Figure 3. There was no correlation between L-arg and RE (R^2^ = 0.037, *p* = 0.348) and between the L-arg/ADMA ratio and RE (R^2^ < 0.001, *p* = 0.92) after 12 weeks of supplementation.

## 4. Discussion

To date, most research has focused on the potential role of omega-3 fatty acids as a vasodilator of the vascular endothelium by increasing nitric oxide (NO) synthesis [25,26,27]. The mechanisms responsible for this phenomenon are not fully understood. However, potential changes in the metabolism of L-arg, ADMA and their metabolites seem to be crucial in understanding these mechanisms. Therefore, in this paper we present for the first time the effect of 12 weeks of supplementation with omega-3 fatty acids in runners on levels of L-arg, ADMA, and their metabolites.

In our study, in response to daily supplementation with 2234 mg of EPA and 916 mg of DHA, we observed an increase in resting plasma L-arg concentration with no change in ADMA concentration. These results are in line with a previous report in non-athletes [28]. As previously mentioned, the mechanism behind this is not fully understood, although it was originally thought that omega-3 fatty acids could decrease plasma ADMA concentrations; however, the evidence for this is scarce and inconsistent. A study with patients with obesity supplemented with EPA and DHA for 8 weeks showed decreased plasma ADMA levels [29]. On the other hand, a study involving trained cyclists showed no changes in plasma ADMA level after three weeks of omega-3 fatty acid supplementation [30]. Other studies have shown that the ADMA level in response to other supplementation interventions is difficult to assess [31,32] due to disturbances resulting from amino acid metabolism/gluconeogenesis and various levels of skeletal muscle damage [33]. Previous studies involving animals [34] and humans [35] identify that it is an increase in L-arg that increases the L-arg/ADMA ratio rather than changes in ADMA concentration; our results are consistent with this. In addition, a higher L-arg/ADMA ratio is positively related to endothelium-dependent vasodilation [36], but this ratio has not previously been used to assess athletes’ exercise capacity. In our previous research we observed improvement in RE in the group supplementing omega-3 fatty acids [7]. In this study, for the first time, according to the authors’ knowledge, the relationships between plasma L-arg, the L-arg/ADMA ratio and RE were investigated. However, increased plasma L-arg levels were not correlated with RE, which is consistent with a study where acute supplementation with 6 g L-arg did not alter oxygen cost of exercise or exercise tolerance in healthy subjects [37]. Nevertheless, these outcomes relate to the acute effect of an increase in plasma L-arginine where NO is rapidly oxidized to its final forms- NO_2_^−^ and NO_3_^−^ [38]. Therefore, it is considered that high levels of L-arg in plasma during resting may be an adaptation of the organism as a result of long-term supplementation with omega-3 fatty acids. While the resting L-arg level is a robust factor influencing the L-arg/ADMA ratio, post-exercise changes in the level of amino acids should be analyzed with caution due to omega-3 fatty acids ability to amplify the effect of exercise [39,40]. Indeed, previous research indicates that 15 min of exercise promotes an increase in L-arg levels in the plasma of athletes [41,42]. Simultaneously, these studies show no changes in ornithine levels after exercise, which is also consistent with our results. Therefore, it seems that the assessment of the level of amino acids (in this case, L-arg and ADMA) after supplementation with omega-3 fatty acids should be performed under resting conditions, which is crucial in the context of studying ergogenic effects. Still, the mechanisms responsible for these changes are the subject of much research, although it is known that omega-3 fatty acids may also act as peroxisome proliferator-activated receptors (PPARs) agonists [43].

The pleiotropic nature of PPARs also includes regulation of the metabolism of amino acids, such as L-arg, thus increasing the bioavailability/synthesis of NO [44]. Interestingly, recent research points to the involvement of omega-3 fatty acids, especially EPA and DHA, in activation of PPARs in rats [45], while omega-3 fatty acids also upregulate PPARγ mRNA expression in blood mononuclear cells in athletes [46]. For this reason, it is believed that PPARγ expression is critical in regulating the metabolism of amino acids such as L-arg. Nevertheless, more research on this topic is needed to understand the changes that occur following omega-3 fatty acid supplementation.

Our study has some limitations. First, the small number of participants means that the observed effects should be treated cautiously. Second, analysis of PPARγ mRNA and protein expression were not performed but would add mechanistic insight into our observations.

## 5. Conclusions

In conclusion, twelve weeks of omega-3 fatty acid supplementation at a dose of 2234 mg of EPA and 916 mg of DHA daily increased plasma L-arg concentration with no change in plasma ADMA levels. The omega-3 intervention promotes an increase in plasma L-arg and the L-arg/ADMA ratio, which indirectly indicates increased bioavailability/NO synthesis. However, our results do not support the relevance of the L-arg/ADMA ratio as a factor improving running economy in male amateur endurance athletes.

## Figures and Tables

**Figure 1 nutrients-14-04749-f001:**
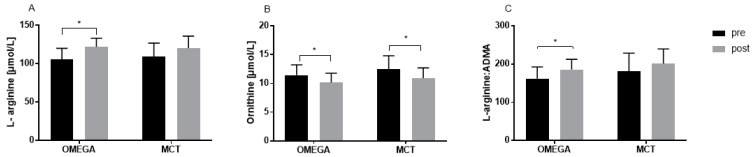
Resting plasma L-arginine (**A**) and ornithine (**B**) levels and L-arginine/ADMA ratios (**C**) pre- and post-12 weeks of supplementation (* *p* < 0.05- pre vs. post).

**Figure 2 nutrients-14-04749-f002:**
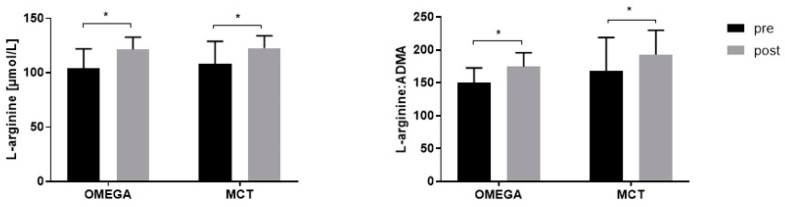
Post-exercise plasma L-arginine levels and L-arginine/ADMA ratios pre- and post-12 weeks of supplementation (* *p* < 0.05- pre vs. post).

**Figure 3 nutrients-14-04749-f003:**
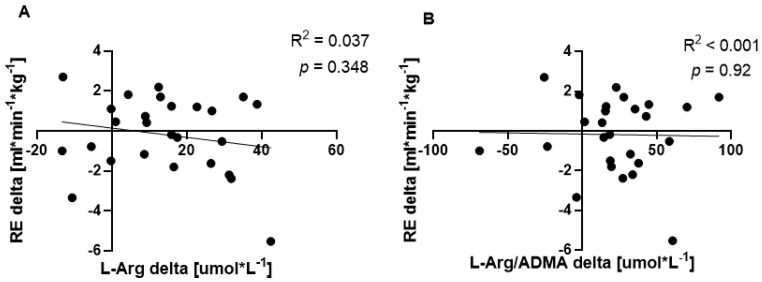
Correlation between resting plasma L-arginine levels (**A**) and L-arginine/ADMA ratios (**B**) and running economy.

**Table 1 nutrients-14-04749-t001:** Characteristics of participants.

Variable	MCT (*n* = 12) Mean ± SD		OMEGA (*n* = 14) Mean ± SD	
Age (years)	37 ± 4		37 ± 3	
Body mass (kg)	78 ± 8		76 ± 11	
Height (cm)	180 ± 4		181 ± 7	
VO_2peak_ (mL*kg^−1^*min^−1^)	54.7 ± 7		53.6 ± 4	
RE (mL*kg^−1^*min^−1^)	Pre	47.7 ± 3.3	Pre	47.6 ± 1.8
	Post	48.7 ± 2.9	Post	46.5 ± 2.4 ^†^
EPA (% of total RBC fatty acids)	Pre	1.2 ± 0.3	Pre	1.1 ± 0.4
	Post	1.2 ± 0.3	Post	4.9 ± 1.1 *^†^
DHA (% of total RBC fatty acids)	Pre	4.4 ± 1.1	Pre	4.7 ± 1.0
	Post	4.5 ± 0.8	Post	6.7 ± 0.8 *^†^
O3I	Pre	5.6 ± 1.4	Pre	5.8 ± 1.3
	Post	5.6 ± 1.1	Post	11.6 ± 1.7 *^†^
Test duration (min: s)	Pre	1091 ± 144	Pre	1111 ± 70
	Post	1137 ± 84 *	Post	1138 ± 85

* *p* < 0.05 post vs. pre; ^†^
*p* < 0.05 MCT vs. OMEGA; SD—standard deviation; EPA—eicosapentaenoic acid; DHA—docosahexaenoic acid; RBC—red blood cell; O3I—Omega-3 index.

**Table 2 nutrients-14-04749-t002:** The effect of 12-week omega-3 fatty acid supplementation on resting plasma levels of L-arginine and its metabolites.

	MCT (*n* = 12) Mean ± SD	OMEGA (*n* = 14) Mean ± SD	Diff	95% CI		*p*
				Lower	Upper	
L-arginine (µmol/L)						
Before	109.4 ± 17.53	105.4 ± 14.67	−4.003	−17.4	9.394	0.744
After	120.4 ± 15.55	122.0 ± 11.12	1.621	−11.78	15.02	0.952
Change	11.00 ± 17.21	16.63 ± 14.87				
*p*	0.109	0.001				
ADMA (µmol/L)						
Before	0.618 ± 0.082	0.669 ± 0.147	0.051	−0.059	0.161	0.496
After	0.611 ± 0.095	0.673 ± 0.139	0.062	−0.482	0.172	0.360
Change	−0.007 ± 0.086	0.004 ± 0.054				
*p*	0.883	0.819				
SDMA (µmol/L)						
Before	0.255 ± 0.03	0.262 ± 0.036	0.007	−0.025	0.04	0.851
After	0.259 ± 0.038	0.264 ± 0.038	0.004	−0.028	0.037	0.940
Change	0.004 ± 0.031	0.001 ± 0.031				
*p*	0.963	0.868				
DMA (µmol/L)						
Before	1.334 ± 0.148	1.301 ± 0.241	−0.033	−0.267	0.202	0.937
After	1.361 ± 0.275	1.394 ± 0.325	0.033	−0.200	0.268	0.934
Change	0.027 ± 0.336	0.092 ± 0.314				
*p*	0.865	0.509				
L-citrulline (µmol/L)						
Before	33.73 ± 6.184	34.97 ± 9.323	1.237	−5.842	8.315	0.903
After	35.36 ± 7.092	33.8 ± 7.905	−1.553	−8.632	5.526	0.852
Change	1.626 ± 3.268	−1.164 ± 3.736				
*p*	0.113	0.265				
Ornithine (µmol/L)						
Before	12.49 ± 2.314	11.45 ± 1.771	−1.048	−2.744	0.649	0.295
After	10.91 ± 1.773	10.17 ± 1.598	−0.740	−2.437	0.956	0.536
Change	−1.582 ± 1.857	−1.274 ± 0.991				
*p*	0.007	<0.001				
L-Arginine:ADMA						
Before	180.9 ± 47.61	162.1 ± 30.45	−18.84	−51.52	13.85	0.343
After	201.5 ± 38.18	185.7 ± 26.54	−15.73	−48.42	16.95	0.470
Change	20.56 ± 41.54	23.66 ± 23.48				
*p*	0.077	0.005				

**Table 3 nutrients-14-04749-t003:** The effect of 12-week omega-3 fatty acid supplementation on post-exercise plasma levels of L-arginine and its metabolites.

	MCT (*n* = 12) Mean ± SD	OMEGA (*n* = 14) Mean ± SD	Diff	95% CI		*p*
				Lower	Upper	
L-arginine (µmol/L)						
Before	108.1 ± 20.8	104.3 ± 17.67	−3.809	−18.1	10.49	0.790
After	122.7 ± 11.41	121.5 ± 11.24	−1.157	−15.45	13.14	0.978
Change	14.55 ± 17.71	17.20 ± 13.75				
*p*	0.016	<0.001				
ADMA (µmol/L)						
Before	0.663 ± 0.095	0.701 ± 0.139	0.038	−0.0611	0.137	0.615
After	0.65 ± 0.089	0.706 ± 0.102	0.056	−0.043	0.155	0.361
Change	−0.013 ± 0.078	0.004 ± 0.064				
*p*	0.566	0.797				
SDMA (µmol/L)						
Before	0.256 ± 0.03	0.272 ± 0.045	0.016	−0.019	0.051	0.489
After	0.265 ± 0.035	0.28 ± 0.039	0.015	−0.02	0.05	0.545
Change	0.009 ± 0.034	0.008 ± 0.032				
*p*	0.374	0.381				
DMA (µmol/L)						
Before	1.505 ± 0.213	1.593 ± 0.374	0.088	−0.249	0.425	0.797
After	1.628 ± 0.373	1.742 ± 0.461	0.115	−0.222	0.452	0.682
Change	0.123 ± 0.341	0.149 ± 0.462				
*p*	0.338	0.248				
L-citrulline (µmol/L)						
Before	34.69 ± 9.013	34.65 ± 11.18	−0.046	−8.486	8.394	>0.999
After	36.98 ± 7.893	34.17 ± 8.511	−2.813	−11.25	5.627	0.693
Change	2.288 ± 3.382	−0.479 ± 4.157				
*p*	0.052	0.952				
Ornithine (µmol/L)						
Before	13.18 ± 2.459	12.25 ± 1.754	−0.932	−2.564	0.07	0.35
After	11.66 ± 1.38	11.78 ± 1.456	0.117	−1.516	1.75	0.983
Change	−1.52 ± 2.546	−0.471 ± 1.497				
*p*	0.063	0.26				
L-Arginine:ADMA						
Before	167.5 ± 51.38	150.8 ± 22.14	−16.78	−47.92	14.35	0.391
After	192.9 ± 37.03	174.6 ± 21.33	−18.33	−49.47	12.8	0.328
Change	25.35 ± 42.21	23.8 ± 17.42				
*p*	0.021	<0.001				

## Data Availability

The data presented in this study are available on request from the corresponding author.

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
