# Peer review of "Increased Plasma L-Arginine Levels and L-Arginine/ADMA Ratios after Twelve Weeks of Omega-3 Fatty Acid Supplementation in Amateur Male Endurance Runners"

_nutrients, 2022, doi:10.3390/nu14224749_

Round 1
Reviewer 1 Report
The manuscript by Jost et al. aimed to illuminate the effects of consuming omega-3 fatty acids on the plasma levels of L-arg, ADMA, and L-arg/ADMA ratio in amateur male endurance runners, and how these changes were associated to their running economy (RE). Such novel findings will potentially hold a great premise to benefit not only male runners, but also a broader population.
Here lists a few of my concerns that could improve the overall quality of the manuscript if being addressed appropriately.
Q1: I would recommend removing the tables (Table 1, Table 2, Table 3) from the main context and keep them as the supplement information. This will the whole manuscript more concise and clearer.
Q2: As for Figure 1 and Figure 2, I would recommend turning all the measurements from Table 2 and Table 3 into bar graphs to show the changed and unchanged results. I believe both changed and unchanged results are important and unique findings generated from this manuscript and will give the readers a complete story. Particularly, I would encourage the authors to annotate the p values that are slightly higher than 0.05 in the bar graphs, such as the MCT L-Arginine:ADMA (p=0.077, above Line 177), MCT L-citrulline (p=0.052, above Line 181) and MCT Ornithine (p=0.063, above Line 181). Along with these potential new findings, the authors should also consider describing in the Results and interpreting these changes in the Discussion.
Q3: As for Figure 3, I would recommend the authors to graph more correlations between the metabolites and RE, such as ADMA, SDMA, DMA, L-citrulline, Ornithine. Accordingly, the authors should consider describing them in the Results and interpreting these in the Discussion.
Q4: Missing figure legend in all figures (Figure 1, Figure 2, Figure 3). The authors should elaborate more details in the figure legend besides a title. Full names should also be given to annotate the acronyms presented in the graphs.
Q5: What’s the application and significance of this study? How does the findings generated from this study potentially benefit humans, not only amateur male runners, but also a broader population? The authors should clearly address how their findings can be applied and utilized to guide future studies. This should be incorporated into the Discussion.
Author Response
Thank you for your comments. Our responses are available in the attachment.
All the best

Reviewer 2 Report
The authors propose a randomized clinical trial with a parallel design to evaluate the effect of long-chain polyunsaturated fatty acids of the 3 series (EPA and DHA) on the levels of Arginine, Ornithine, symmetric and asymmetric dimethylarginine. These molecules seem to represent the junction between PUFAs and the effect on recovery and physical performance mediated by nitric oxide.
26 amateur athletes were recruited for intervention with PUFAs or MCTs (placebo) for 12 weeks. Erythrocyte levels of PUFAs and arginine metabolites studied were detected at baseline and the end of the intervention period and both pre and postworkout. The increase in erythrocyte polyunsaturated fatty acids was highlighted, as confirmed by the initial study already published. Arginine levels were increased in the intervention group at resting but not in the post-workout or placebo group. Furthermore, the Arginine:ADMA ratio was increased at rest in the intervention group but not in the placebo. Post-workout Arginine levels and Arg:ADMA ratio increased in both groups. Although RE improved in the intervention group, no correlation was found with Arg or Arg:ADMA.
The manuscript is well structured, fluent and with adequate bibliographic references. The materials are adequately described and the experimental design seems adequate. I have just a few suggestions, in hopes of implementing the manuscript:
- Even if among the limits the authors have highlighted the number of participants, it could be useful to specify the power analysis
- Although the relationship between Arginine and metabolites and nitric oxide on performance and recovery is explained, there is only a final hint about the possible link between PUFAs and NO in the final discussion. A molecular/cellular hint in the introduction on the possible link proposed between PUFAs and Arginine would be useful
- In the introduction and abstract, I would clarify better that the association between RE and PUFAs highlighted by the authors in the study refers only compared with placebo and not between pre and post-intervention. Similarly, in the abstract, the lack of association with RE refers to the metabolites of arginine and not to the levels of polyunsaturated fatty acids
Author Response
Thank you for your comments. Our responses are available in the attachment.
All the best.
